# Optimal Configuration of Electrochemical Energy Storage for Renewable Energy Accommodation Based on Operation Strategy of Pumped Storage Hydro

**Linjun Shi** [1,*]**, Fan Yang** [1,*]**, Yang Li** [1]**, Tao Zheng** [2]**, Feng Wu** [1] **and Kwang Y. Lee** [3]

[1] College of Energy and Electrical Engineering, Hohai University, Nanjing 210098, China
[2] State Key Laboratory of Smart Grid Protection and Control, Nari Group Corporation, Nanjing 211106, China
[3] Department of Electrical & Computer Engineering, Baylor University, Waco, TX 76798, USA
**\*** Correspondence: 19990041@hhu.edu.cn (L.S.); yf199808@163.com (F.Y.)

**Abstract:** Due to the volatility of renewable energy resources (RES) and the lag of power grid construction, grid integration of large-scale RES will lead to the curtailment of wind and photovoltaic power. Pumped storage hydro (PSH) and electrochemical energy storage (EES), as common energy storage, have unique advantages in accommodating renewable energy. This paper studies the optimal configuration of EES considering the optimal operation strategy of PSH, reducing the curtailment of wind and photovoltaic power in the power grid through the cooperative work of PSH and EES. First, based on the curtailment of RES, with the goal of improving the accommodation of RES, a combined operation optimization model of PSH and EES is proposed. Then, an optimal configuration method of EES capacity is proposed to meet the power curtailment rate in the power grid. Finally, the simulation is carried out in the actual power grid and the CPLEX solver is used to solve the optimization, and the rationality and economy of the optimization are analyzed and discussed. The simulation results show that, based on the combined operation of PSH and EES, by rationally configuring the capacity of EES, the desired power curtailment rate of the power grid can be achieved, and the necessity of configuring variable speed units is verified.

**Keywords:** pumped storage hydro (PSH); electrochemical energy storage (EES); renewable energy accommodation; power curtailment; operation strategy; capacity configuration; variable speed unit

## 1. Introduction

In recent years, in response to increasingly serious environmental problems, many countries have proposed renewable energy development goals to promote the transformation of the energy consumption structure. Wind power generation and photovoltaic (PV) power generation, as relatively mature renewable energy power generation technologies, have received extensive attention [1,2]. However, when wind power and PV power generation replace fossil-fuel power, due to their randomness, volatility and intermittency, they will have a greater impact, which may open spatial and temporal gaps between the availability of the energy and its consumption by the end-users [3]. Prioritizing the accommodation of wind power and PV power reduces the participation of conventional power sources (such as thermal power) in the power grid and, moreover, increases the output volatility of conventional power sources [4–6]. When these outputs cannot be coordinated and balanced with each other, it will lead to the curtailment of wind power and PV power. How to ensure the safe and stable operation of the power grid while accommodating wind, PV and other clean energy sources, has become a major challenge. In order to address these issues, it is necessary to develop suitable energy storage systems for the power grid.

As a means of transferring electrical energy in real time, energy storage devices can compensate for the volatility and randomness, and accommodate large-scale renewable energy, thereby reducing wind and PV power curtailment [7–9]. The pumped storage

hydro (PSH) plant is currently a relatively mature large-scale energy storage device, which has various functions such as peak shaving, frequency modulation, phase modulation and spinning reserve [10,11]. The coordinated operation of PSH and renewable energy can improve the peak shaving capability of the system, which plays an important role in reducing generation curtailment and improving the utilization rate of renewable energy [12,13]. However, PSH is limited by the pump turbines, which must frequently change operating conditions to maintain the stability of the power grid. As the conversion process is a transient process, during this period, high-amplitude pressure pulsation, which threatens the safe and stable operation, will appear as well [14,15].

As PSH is limited by geographical location and pump turbines, electrochemical energy storage (EES) is an alternative large-scale storage device, which is more flexible and can also transfer wind and PV power [16,17]. Moreover, EES has a fast response and flexible charging and discharging characteristics, which can make up for the shortcomings of the slow response of traditional PSH units [18–20]. Suitable technologies of EES can be sodium-ion batteries [21], potassium-ion batteries [22], flow batteries [23] and special/safe types of lithium-ion batteries [24]. Therefore, the development of synergistic services between PSH and EES is of great significance for further improving the accommodation rate of renewable energy and ensuring the safe and stable operation of the power grid.

Under the background of renewable energy accommodation, there have been increasing studies on the optimal scheduling and related configuration issues of PSH and EES. When it comes to the study on energy storage for renewable energy accommodation, the existing literature mostly starts with the renewable energy accommodation by either PSH or EES alone. Referring to PSH for renewable energy accommodation, Ma et al. [25] introduced PSH into a microgrid containing wind and PV power and used PSH to compensate for the intermittent nature of renewable energy, thus verifying that PSH can achieve energy autonomy in remote communities. Considering the uncertainty of wind energy in time and space, Xia et al. [26] established a unit combination model with PSH to coordinate the short-term dispatch of wind power, and the simulation verified that the method can be applied to large-scale power systems. Lin et al. [27] considered the benefits of PSH and the economics of thermal power deep peak shaving, proposing an optimal scheduling model for combined PSH and thermal power peak shaving, and verified that the scheme can greatly improve the economy.

In addition, when it comes to renewable energy accommodation of EES, Dui et al. [28] introduced EES into wind farms and used its peak shaving and wind power forecast error compensation to reduce wind power curtailment. They also considered comprehensively the operational constraints of the power system and the uncertainty of wind power to optimize the capacity of EES installed in wind farms. Zhang et al. [29] combined the reactive power compensation of the PV inverter with the active power curtailment of EES and realized the improvement of the voltage curve and the reduction of the power curtailment while meeting the necessary load demand. Ye et al. [30] built a joint system of thermal power units and EES to participate in power grid peak regulation and renewable energy accommodation, considering the difficulty of grid peak regulation and the overall economy. They then proposed a hierarchical optimal dispatch model based on the optimal energy abandonment rate. It can be seen from these investigations that few study the synergistic accommodation of renewable energy by PSH and EES and discuss the need of configuring variable speed for PSH power units.

In order to improve the accommodation of renewable energy, this paper studies the synergistic operation of PSH and EES and develops the optimal configuration of EES, aiming at the shortcomings of the single energy storage form. Firstly, a combined operation optimization model of PSH and EES is formulated with the goal of improving renewable energy accommodation. Then, based on rationally arranging the operation of PSH, capacity optimization of EES is performed to meet the requirements of the power curtailment rate in the grid. Finally, based on the operation strategies of PSH in different scenarios, considering the participation of variable speed units, its influence on the optimization effect

and operation economy is explored. Through the case study of the actual power grid, the dispatching strategies of PSH and EES are obtained based on the optimization model, and the configuration recommendations of EES are obtained based on the curtailment rate, thus verifying the effectiveness of the method. In addition, the discussion on the variable speed unit verifies its advantages in accurate compensation and provides new possibilities for the construction of pumped storage power stations in the future.

## 2. Operation Optimization Model of PSH

### 2.1. Power Curtailment and Objective Function after PSH Optimization

During the actual operation of the power grid, the balance between the generated power and the load demand should be maintained at every moment. Since the power generation of renewable energy is affected by external weather conditions, its output fluctuates greatly. Therefore, under normal circumstances, the dispatching center of the power grid will formulate corresponding operation strategies to increase the renewable energy power generation as much as possible under the premise of ensuring the safe operation of the power grid. However, the phenomenon of the power curtailment of wind and PV power will inevitably occur in the actual operation.

Considering factors such as the output characteristics of renewable energy and the transmission capacity of the power grid in the actual power grid, with the real-time power generation of wind and PV power as input, the power curtailment in each period of the power grid is optimized in the simulation. The initial total amount of curtailed power in the power grid is shown in the following formula:

$$W_{C,A} = \sum_{t=1}^{T} P_{c,t} \Delta t \tag{1}$$

where $W_{C,A}$ is the total amount of curtailed power before optimization, $P_{c,t}$ is the curtailed power in the $t$-th period, $T$ is the total number of periods in the simulation cycle, and $\Delta t$ is the length of the period.

By adding the PSH power station, renewable energy accommodation can be effectively realized, which is embodied in pumping water to store electric energy during the period of power curtailment, and electricity can be generated to release electric energy during the remaining periods when there is room for accommodation. It should be noted that if there is already power curtailment during this period, the power generation of PSH will exacerbate the power curtailment, and the power curtailment caused needs to be enforced. Therefore, the total amount of curtailed power after the participation of PSH is as follows:

$$W_{C,B}' = \sum_{t=1}^{T} \left( P_{c,t} - \sum_{k=1}^{K} P_{k,t}^{\mathrm{hp}} + Y_P \sum_{k=1}^{K} P_{k,t}^{\mathrm{hg}} \right) \Delta t \tag{2}$$

where $W_{C,B}'$ is the preliminary total amount of curtailed power after the PSH optimization; $P_{k,t}^{\mathrm{hp}}$ and $P_{k,t}^{\mathrm{hg}}$ are the power of the $k$-th unit under pumping and generating conditions in the $t$-th period, respectively; $K$ is the total number of units; $Y_P$ is a Boolean variable indicating whether there is power curtailment in this period, taking 1 to indicate yes, and taking 0 to indicate no.

Since the curtailed power after optimization will not be less than 0, the part of the curtailed power that is less than 0 after the PSH compensation is set to 0, and the final total amount of curtailed power after PSH optimization is:

$$W_{C,B} = \max\left(W_{C,B}', 0\right) \tag{3}$$

where $W_{C,B}$ is the final total amount of curtailed power after PSH optimization.

Therefore, knowing the initial power curtailment and parameters of PSH units, by reasonably arranging the output of PSH units, the total amount of curtailed power is reduced as much as possible, and the goal is to minimize the total amount of curtailed power after adding PSH optimization. The objective function is as follows:

$$\min F_1 = W_{C,B} \tag{4}$$

where $F_1$ is the final total amount of curtailed power after PSH optimization.

### 2.2. Constraints of PSH

#### 2.2.1. Power Constraints of PSH Units

Under pumping and generating conditions, the operating power of each PSH unit is between certain upper and lower limits:

$$y_{k,t}^{\mathrm{p}} P_{\mathrm{hp,min}} \leq P_{k,t}^{\mathrm{hp}} \leq y_{k,t}^{\mathrm{p}} P_{\mathrm{hp,max}} \tag{5}$$

$$y_{k,t}^{\mathrm{g}} P_{\mathrm{hg,min}} \leq P_{k,t}^{\mathrm{hg}} \leq y_{k,t}^{\mathrm{g}} P_{\mathrm{hg,max}} \tag{6}$$

where $P_{\mathrm{hp,max}}$ and $P_{\mathrm{hp,min}}$ are, respectively, the upper and lower limits of the pumping power of the unit; $P_{\mathrm{hg,max}}$ and $P_{\mathrm{hg,min}}$ are, respectively, the upper and lower limits of the generating power of the unit; $y_{k,t}^{\mathrm{p}}$ and $y_{k,t}^{\mathrm{g}}$ are Boolean variables that, respectively, represent whether the units are in the pumping condition and the generating condition, taking 1 to indicate yes, and taking 0 to indicate no.

#### 2.2.2. Single Working Condition Constraints of PSH Units and Stations

In the case of the same unit or the same power station, the Boolean variable representing the pumping or generating condition is set to ensure the single working condition of the unit or power station. The sum of the two variables is less than or equal to 1, indicating that the working conditions of the unit or power station are consistent at the same time:

$$y_{k,t}^{\mathrm{p}} + y_{k,t}^{\mathrm{g}} \leq 1 \tag{7}$$

$$Y_{\mathrm{P}} + Y_{\mathrm{G}} \leq 1 \tag{8}$$

where $Y_{\mathrm{P}}$ and $Y_{\mathrm{G}}$ are Boolean variables that, respectively, represent whether the power station is in the pumping condition and the generating condition, taking 1 to indicate yes, and taking 0 to indicate no.

#### 2.2.3. Constraint on the Number of Working Units

Whether in the pumping condition or in the generating condition, the maximum number of working units in the PSH station is $K$, as shown in the following:

$$\sum_{k=1}^{K} y_{k,t}^{\mathrm{p}} \leq K Y_{\mathrm{P}} \tag{9}$$

$$\sum_{k=1}^{K} y_{k,t}^{\mathrm{g}} \leq K Y_{\mathrm{G}} \tag{10}$$

#### 2.2.4. Constraints on Water Level of Power Station Reservoir and Its Fluctuation

The real-time water level of the power station reservoir is between certain upper and lower limits, and it must be ensured that the water level at the beginning and end of a dispatch cycle is consistent. By setting the water-to-electricity conversion coefficients $\eta_{\mathrm{p}}$ and $\eta_{\mathrm{g}}$ under the pumping and generating conditions, respectively, the water level of the reservoir and the power of the power station are linked:

$$E_{\mathrm{h,min}} \leq E_{\mathrm{h,t}} \leq E_{\mathrm{h,max}} \tag{11}$$

$$E_{\mathrm{h},t+1} = E_{\mathrm{h},t} + \left( \eta_{\mathrm{P}} \sum_{k=1}^{K} P_{k,t}^{\mathrm{hp}} - \eta_{\mathrm{g}} \sum_{k=1}^{K} P_{k,t}^{\mathrm{hg}} \right) \Delta t \tag{12}$$

$$E_{\mathrm{h},t_0} = E_{\mathrm{h},t_{\mathrm{end}}} \tag{13}$$

where $E_{\mathrm{h},t}$ is the real-time water level of the upper reservoir of the power station in the *t*-th period; $E_{\mathrm{h,max}}$ and $E_{\mathrm{h,min}}$ are, respectively, the upper and lower limits of the water level of the upper reservoir of the power station; $t_0$ and $t_{\mathrm{end}}$ are, respectively, the beginning and end of the dispatching cycle. The storage capacity constraint on the upper reservoir also represents the lower reservoir constraint.

2.2.5. Constraint on the Maximum Number of Start-Stop Switching Times of the Units

The maximum number of start-stop switching times are used to limit the start and stop of the unit to avoid frequent switching of the unit, thereby reducing the start-up and shut-down cost of PSH, which can be expressed as follows:

$$\sum_{t=2}^{T} \left| y_{k,t}^{\mathrm{p}} - y_{k,t-1}^{\mathrm{p}} \right| \leq n_{\mathrm{p}} \tag{14}$$

$$\sum_{t=2}^{T} \left| y_{k,t}^{\mathrm{g}} - y_{k,t-1}^{\mathrm{g}} \right| \leq n_{\mathrm{g}} \tag{15}$$

where $n_{\mathrm{p}}$ and $n_{\mathrm{g}}$ are, respectively, the number of start-stop switching times of the unit under the pumping and generating conditions in the dispatching cycle.

## 3. Operation and Configuration Optimization Model of EES

### 3.1. Power Curtailment after EES Optimization

On the basis of the operation optimization of the PSH station, EES can be configured to realize further accommodation of renewable energy, which is embodied in the charging and storing of electric energy during the power curtailment period, and discharging and releasing of electric energy during the rest period. It should be noted that if there is already power curtailment during this period, the discharge of EES will also exacerbate the power curtailment, so it is necessary to enforce the power curtailment. Therefore, the total amount of curtailed power after EES optimization is as follows:

$$W_{\mathrm{C,C}}{}' = W_{\mathrm{C,B}}{}' - \sum_{t=1}^{T} (P_{\mathrm{ch},t} - Y_{\mathrm{E}} P_{\mathrm{dis},t}) \Delta t \tag{16}$$

where $W_{\mathrm{C,C}}{}'$ is the preliminary total amount of curtailed power after EES optimization; $P_{\mathrm{ch},t}$ and $P_{\mathrm{dis},t}$ are, respectively, the charging and discharging power of EES in the *t*-th period; $Y_{\mathrm{E}}$ is a Boolean variable indicating whether there is power curtailment in this period, taking 1 to indicate yes, and taking 0 to indicate no.

Since the curtailed power after optimization will not be less than 0, the part of the curtailed power that is less than 0 after the EES compensation is set to 0, and the final total amount of curtailed power after EES optimization is:

$$W_{\mathrm{C,C}} = \max \left( W_{\mathrm{C,C}}{}', 0 \right) \tag{17}$$

where $W_{\mathrm{C,C}}$ is the final total amount of curtailed power after EES optimization.

### 3.2. Objective Function of EES Operation Optimization

After the operation of PSH is optimized, based on the curtailed power after PSH optimization, the selection and parameters of EES are known, and the operation of EES is also reasonably arranged with the goal of minimizing the curtailed power. The synergy between PSH and EES can further improve the optimization effect of the curtailment of wind and PV power. The objective function of EES operation optimization is as follows:

$$\min F_2 = W_{\text{C,C}} \tag{18}$$

where $F_2$ is the total amount of curtailed power after PSH and EES optimization.

### 3.3. Objective Function of EES Configuration Optimization

　　Similarly, after the operation of PSH is optimized, the power curtailment and the parameters of EES are known, the rated power and rated capacity of EES are reasonably configured, and the charging and discharging power of EES is optimized. By reducing the total amount of power curtailment after optimization as much as possible, it can reach the expected power curtailment rate set in advance. With the goal of achieving the minimum rated power of EES required to achieve the expected power curtailment rate, the EES capacity configuration at this time is solved. Since there is a fixed multiple relationship between the rated capacity and rated power of EES in this paper, the rated capacity can be obtained when the rated power is solved. The objective function of EES configuration optimization is as follows:

$$\min F_3 = P_{\text{ESS}} \tag{19}$$

where $F_3$ is the rated power of EES.

### 3.4. Constraints of EES
#### 3.4.1. Power Constraints of EES

　　Considering that excessive charging and discharging current will damage the performance of EES, the real-time operating power of EES cannot exceed the rated power:

$$0 \leq P_{\text{ch},t} \leq y_{\text{ch},t} P_{\text{ESS}} \tag{20}$$

$$0 \leq P_{\text{dis},t} \leq y_{\text{dis},t} P_{\text{ESS}} \tag{21}$$

where $y_{\text{ch},t}$ and $y_{\text{dis},t}$ are Boolean variables that characterize the charge and discharge states of EES, respectively, and satisfy $y_{\text{ch},t} + y_{\text{dis},t} \leq 1$ at any time.

#### 3.4.2. SOC Constraints of EES

　　The ratio of the remaining capacity of EES to its full charging capacity is the state of charge (SOC). The remaining capacity of EES must meet certain constraints during operation, that is, the SOC of EES must be between certain upper and lower limits. In addition, the consistent SOC at the beginning and end of each scheduling cycle of EES can ensure the periodicity of its continuous operation:

$$SOC(t) = SOC(t-1) + \frac{[\eta_{\text{ch}} P_{\text{ch},t} - P_{\text{dis},t}/\eta_{\text{dis}}]\Delta t}{E_{\text{ESS}}} \tag{22}$$

$$SOC(t_0) = SOC(t_{\text{end}}) \tag{23}$$

$$SOC_{\min} \leq SOC(t) \leq SOC_{\max} \tag{24}$$

where $SOC(t)$ is the SOC of EES in the $t$-th period; $\eta_{\text{ch}}$ and $\eta_{\text{dis}}$ are the energy conversion efficiencies of EES during charging and discharging, respectively; $SOC_{\max}$ and $SOC_{\min}$ are the upper and lower limits of the SOC of EES, respectively.

#### 3.4.3. Constraint of Expected Power Curtailment Rate

　　The power curtailment rate after optimization should not be greater than the expected power curtailment rate, as shown in the following:

$$\sigma \leq \sigma_0 \tag{25}$$

where $\sigma_0$ is the expected power curtailment rate.

## 4. Operation and Configuration Optimization Algorithms of PSH and EES

The optimization in this paper uses the YALMIP toolbox to call the CPLEX solver to solve. The CPLEX solver has the advantages of fast convergence speed and good robustness to linear problems, and it is suitable for solving the optimization model in this paper [31].

The overall idea of the optimization algorithm is shown in Figure 1, and the detailed steps are as follows:

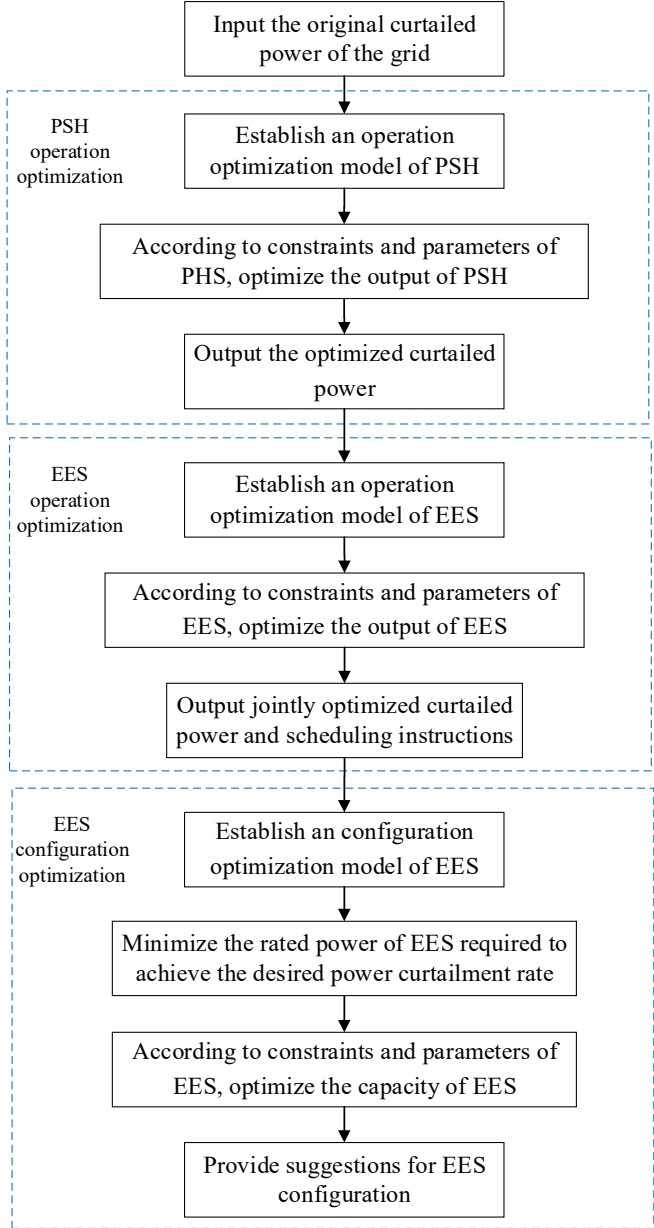

**Figure 1.** Flowchart of the optimization algorithm.

Step 1: Select an appropriate simulation period and input the original curtailed power of the power grid.

Step 2: Establish an operation optimization model of PSH and set constraints with the goal of minimizing the power curtailment.

Step 3: Based on the original curtailed power and parameters of PSH, use the CPLEX solver to solve the output of PSH. The optimized curtailed power is the output to the EES operation optimization model.

Step 4: Establish an operation optimization model of EES, with the goal of minimizing the curtailed power, set constraints, and solve the output of EES.

Step 5: Establish the configuration optimization model of EES, and then set the constraints to achieve the minimum rated power of EES required to achieve the expected power curtailment rate.

Step 6: Based on the curtailed power after the PSH optimization and parameters of EES, set the index of the expected power curtailment rate, and use the CPLEX solver to solve the configuration of EES.

Step 7: Compare and analyze the curtailed power before and after optimization, and then provide suggestions on the scheduling of PSH and EES, as well as the configuration of EES.

## 5. Case Study

### 5.1. Case Parameters

In order to verify the rationality of the above models and algorithms, this paper selects the parameters of PSH and the data of the curtailed power in an actual power grid for the case study.

### 5.1.1. Parameters of PSH

There are four PSH power stations in the power grid of the selected case, and the number of units and the upper and lower limits of the power of each power station are shown in Table 1. The difference between the fixed-speed unit and the variable-speed unit is mainly whether the power under the pumping condition is adjustable, and the power of the fixed-speed unit under the pumping condition is the full power state. This paper first discusses the case where all units are fixed-speed units, and the power range of PSH power generation is 50–100%. The total installed capacity of the four power stations is 6070 MW.

**Table 1.** Parameters of the units of PSH.

| Serial Number of the Power Station | Number of Units | Capacity of a Single Unit (MW) | Total Installed Capacity (MW) |
|---|---|---|---|
| 1 | 3 | 90 | 270 |
| 2 | 4 | 200 | 800 |
| 3 | 12 | 300 | 3600 |
| 4 | 4 | 350 | 1400 |

In this case, the four PSH power stations reach the upper limit of the water level after 6 h of pumping at full power, and the water levels at the beginning and end of the day are the same. The water-to-electricity conversion coefficients under the pumping condition and generating condition are set to be 251.62 and 318.24, respectively, and the unit is $m^3/(MW \cdot h)$ [27]. The start-stop switching times of the pumping and generating conditions are set to be three, that is, to ensure that each PSH power station does not pump and generate more than three times a day.

### 5.1.2. Parameters of Power Curtailment

The data of power curtailment in the case is selected from the actual power grid in an area. The data of power curtailment in a typical week is selected, one hour is a sampling point, and there are 168 sampling points in 7 days. The data of power curtailment is shown in Figure 2.

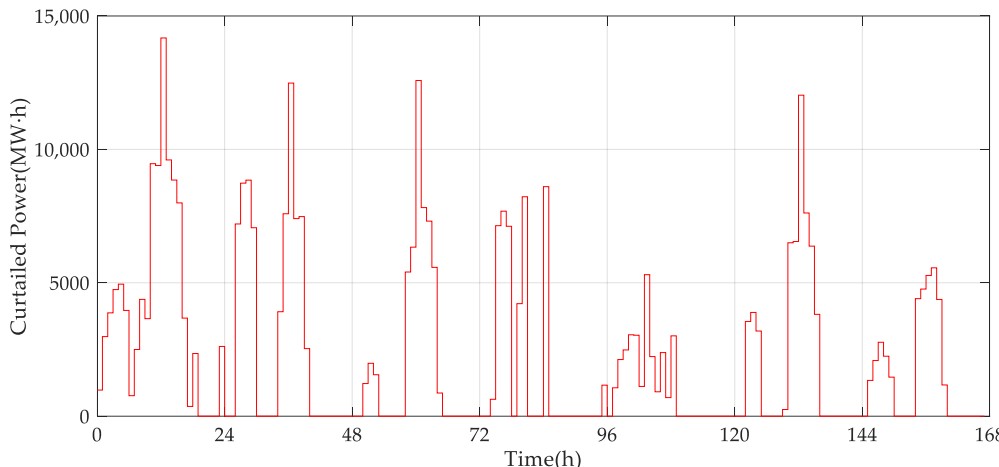

**Figure 2.** Original power curtailment data of the power grid.

### 5.1.3. Parameters of EES

In terms of the selection of EES, this paper directly selects the widely used lithium iron phosphate battery as an example. Its basic parameters are as follows: the ratio of the rated capacity and power of the battery is 2, and the charge-discharge efficiency is 90%. The upper limit of the SOC of the battery is 0.8, and the lower limit is 0.2. The purchase cost of the unit power is 600 yuan/kW, and the investment unit price of the unit capacity is 1600 yuan/(kW·h) [32].

### 5.2. Operation Optimization of PSH and EES

In order to verify the rationality of the above models and algorithms, this paper selects the parameters of PSH and the data of curtailed power in an actual power grid for case study.

For the operation optimization of the PSH power station, according to the objective function of PSH operation optimization and various constraints, the hourly output of PSH in a typical week is calculated, and the operation curve of a typical week is drawn, as shown in Figure 3.

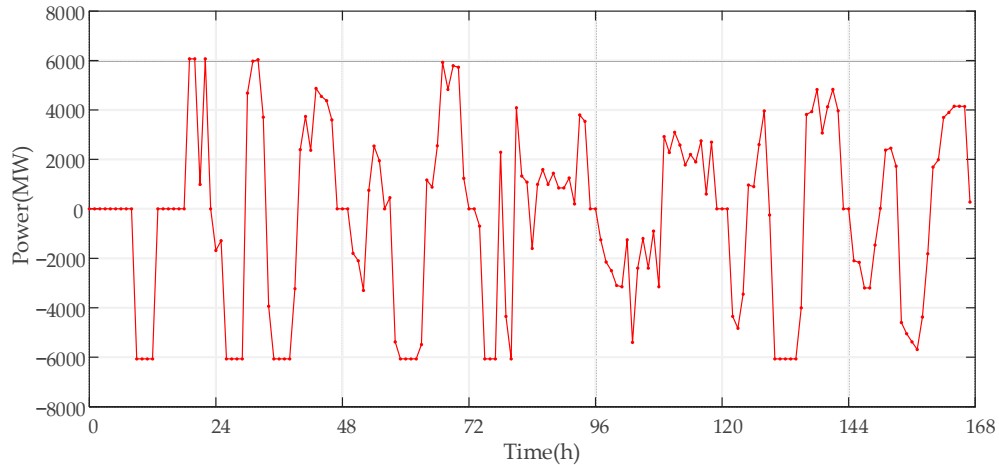

**Figure 3.** Operation curve of PSH in a typical week.

For the operation optimization of EES, according to the objective function of operation optimization and various constraints, the output of EES in a typical week is calculated. Since the configuration optimization of EES has not yet been carried out, the rated power of 2000 MW is directly selected as the assumption for the study of operation optimization. The operation curve of EES in a typical week is shown in Figure 4.

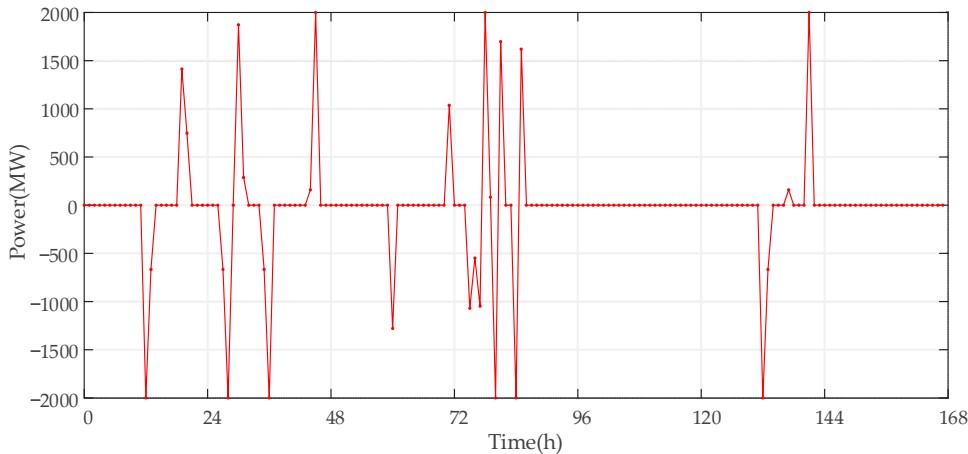

**Figure 4.** Operation curve of EES in a typical week.

The operation optimization results of PSH and EES are jointly analyzed, and the seven days of a typical week are further divided into three representative scenarios for specific analysis according to the pumping and generating (charging and discharging) and accommodation conditions.

### 5.2.1. Scenario 1

The first representative scenario is the simulation result of the second day, and the operation curve of PSH is shown in Figure 5. Among them, the output is negative when pumping water, and positive when generating electricity.

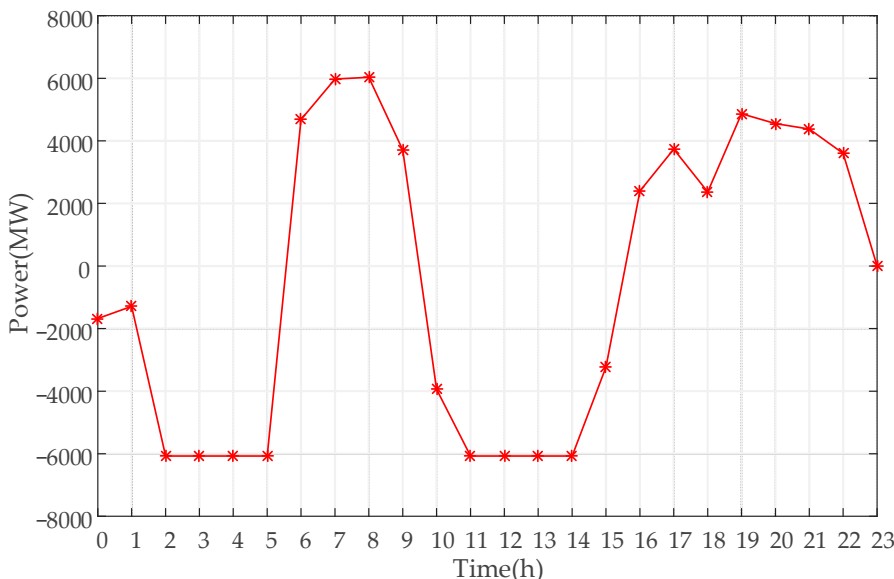

**Figure 5.** Operation curve of PSH (scenario 1).

The operation curve of EES is shown in Figure 6, where the output is negative when charging, and positive when discharging.

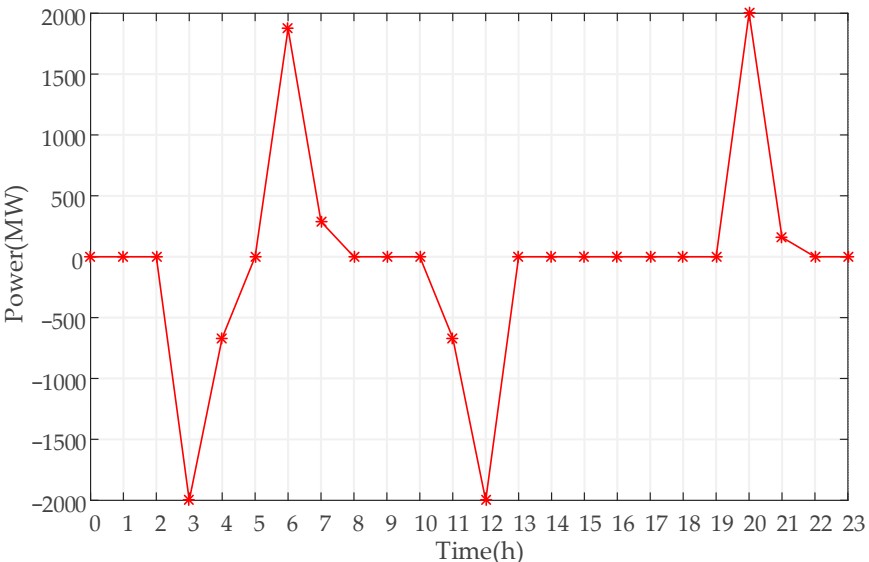

**Figure 6.** Operation curve of EES (scenario 1).

It can be seen from Figures 5 and 6 that the PSH and EES operations in this scenario are pumping twice and generating twice (charging twice and discharging twice).

In this case, the change in the SOC of EES is shown in Figure 7.

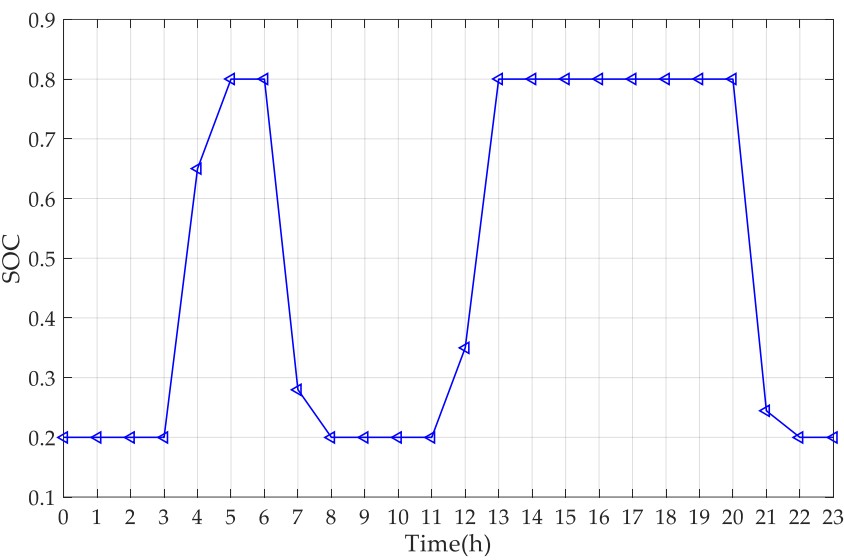

**Figure 7.** SOC curve of EES (scenario 1).

As can be seen from the above figure, the SOC of EES is within a limited range, meeting the requirements of the upper and lower limits of SOC, and the state of the beginning and end of the cycle is consistent.

The curtailed power before and after optimization are compared on a graph, and the scenario is analyzed in detail in relation to the changes in the curtailed power. The comparison of power curtailment is shown in Figure 8.

As can be seen from Figure 8, there is a large amount of power curtailment during certain periods of the day. The power curtailment is significantly reduced through the accommodation effect of PSH. However, due to the limited capacity of the PSH unit, the reduction may be limited. If we want to further reduce the power curtailment, we need to configure the EES.

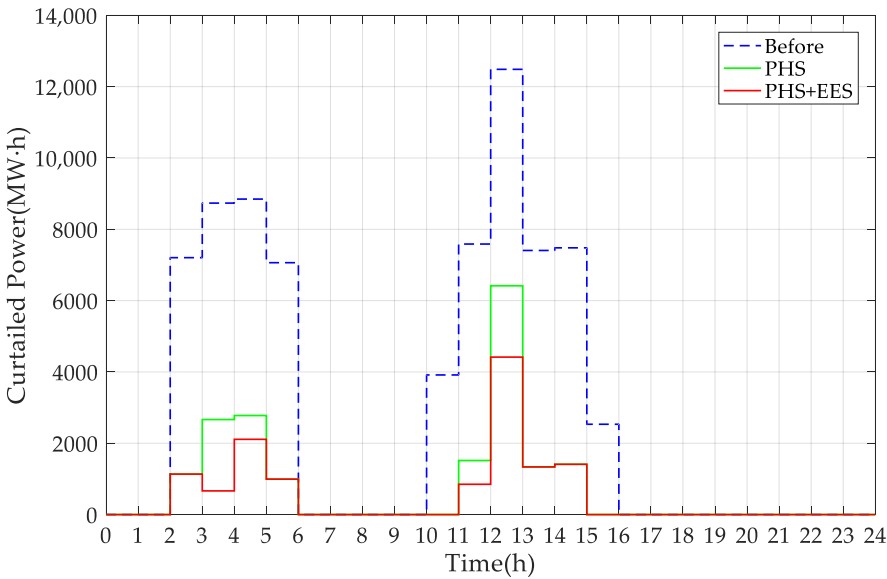

**Figure 8.** Comparison of power curtailment before and after optimization (scenario 1).

After adding the EES, the remaining part of the renewable energy is accommodated by the charging of the EES, which further reduces the power curtailment. Comparing the curtailed power before and after optimization, compared with the accommodation of PSH alone, the curtailed power has a further significant decrease after adding the EES. Therefore, the operation optimization of the PSH and EES is reasonable, which verifies the effectiveness of the operation strategy.

From the analysis of data changes, there is power curtailment from 2:00 to 5:00 and from 10:00 to 17:00 in this typical day. At this time, through the cooperative work of PSH and EES, the power curtailment has dropped significantly. The curtailed power reached the maximum value of 12,581.66 MW·h from 12:00 to 13:00. Due to the pumping of the PSH unit, the curtailed power dropped to 6511.66 MW·h. On this basis, the EES was charged for accommodation, and the curtailed power further decreased to 5232.31 MW·h. The optimization effect after adding the EES is related to the pre-selected EES capacity. If we want to get better optimization effect, we need to configure larger capacity EES. This verifies the feasibility of PSH and EES synergistically accommodating renewable energy.

5.2.2. Scenario 2

The second representative scenario is the simulation result of the 4th day, and the operation curve of PSH is shown in Figure 9.

The operation curve of EES is shown in Figure 10.

It can be seen from Figures 9 and 10 that the PSH and EES operations in this scenario are pumping three times and generating three times (charging three times and discharging three times).

In this case, the change in the SOC of EES is shown in Figure 11.

As can be seen from the above figure, the SOC of EES is within a limited range, meeting the requirements of the upper and lower limits of SOC, and the state of the beginning and end of the cycle is consistent.

The curtailed power before and after optimization are compared on a graph, and the scenario is analyzed in detail in relation to the changes in the curtailed power. The comparison of power curtailment is shown in Figure 12.



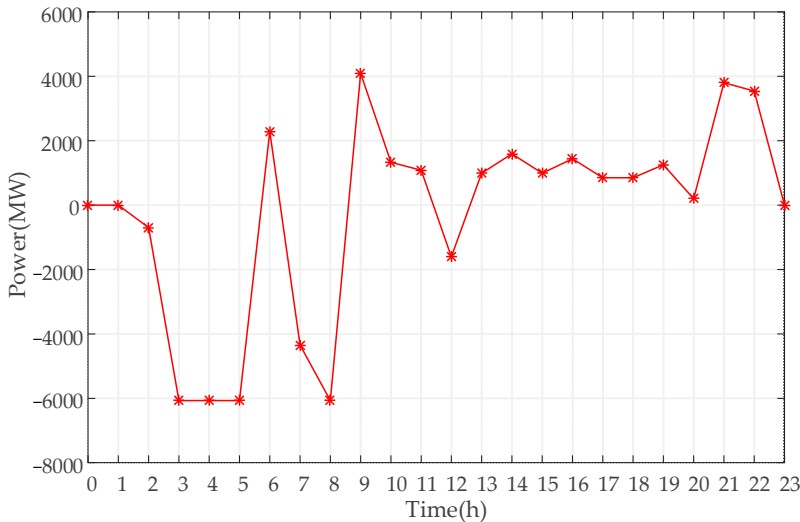

**Figure 9.** Operation curve of PSH (scenario 2).

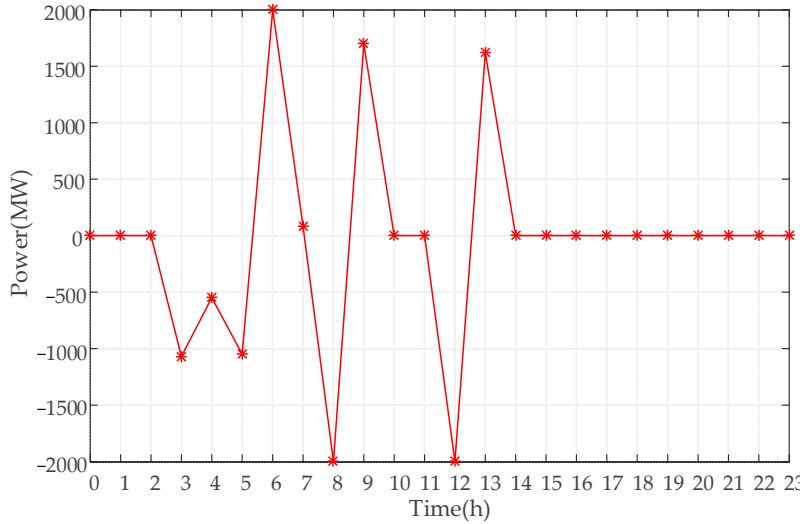

**Figure 10.** Operation curve of EES (scenario 2).

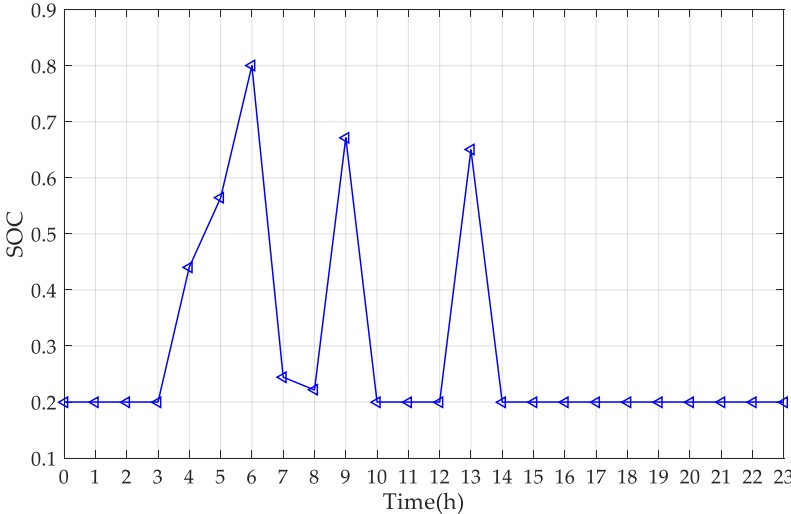

**Figure 11.** SOC curve of EES (scenario 2).

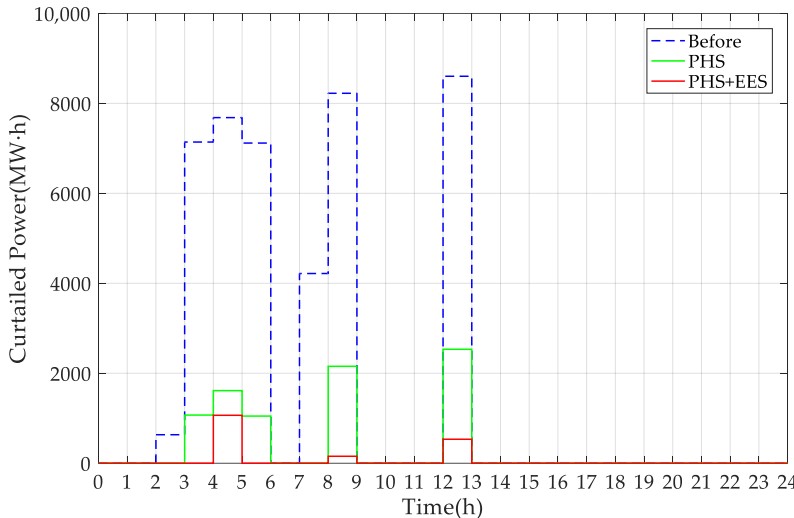

**Figure 12.** Comparison of power curtailment before and after optimization (scenario 2).

From the analysis of data changes, there is power curtailment from 2:00 to 6:00, 7:00 to 9:00, and 12:00 to 13:00 in this typical day. At this time, through the cooperative work of PSH and EES, the power curtailment has dropped significantly. The curtailed power reached the maximum value of 8602.84 MW·h from 12:00 to 13:00. Due to the pumping of the PSH unit, the curtailed power dropped to 2532.84 MW·h. On this basis, the EES was charged for accommodation, and the curtailed power further decreased to 532.84 MW·h.

### 5.2.3. Scenario 3

The third representative scenario is the simulation result of the 6th day, and the operation curve of the PSH is shown in Figure 13.

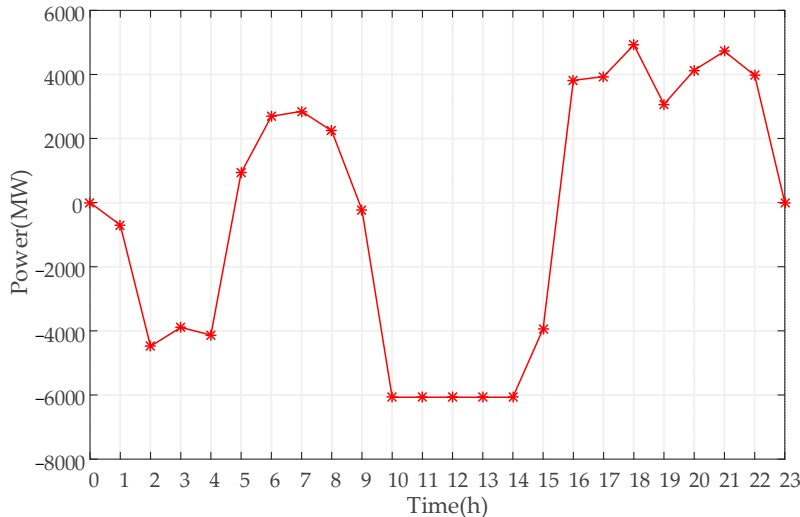

**Figure 13.** Operation curve of PSH (scenario 3).

The operation curve of the EES is shown in Figure 14.

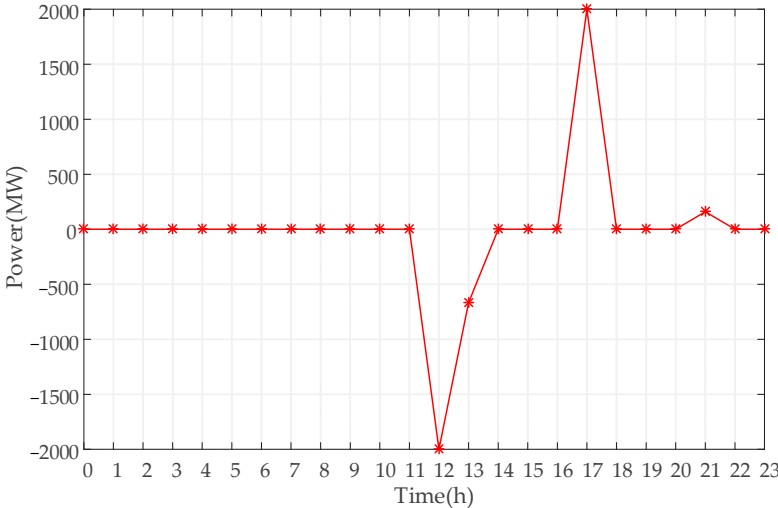

**Figure 14.** Operation curve of EES (scenario 3).

It can be seen from Figures 13 and 14 that the PSH and EES operations in this scenario are pumping twice and generating twice (charging once and discharging once).

In this case, the change in the SOC of EES is shown in Figure 15.

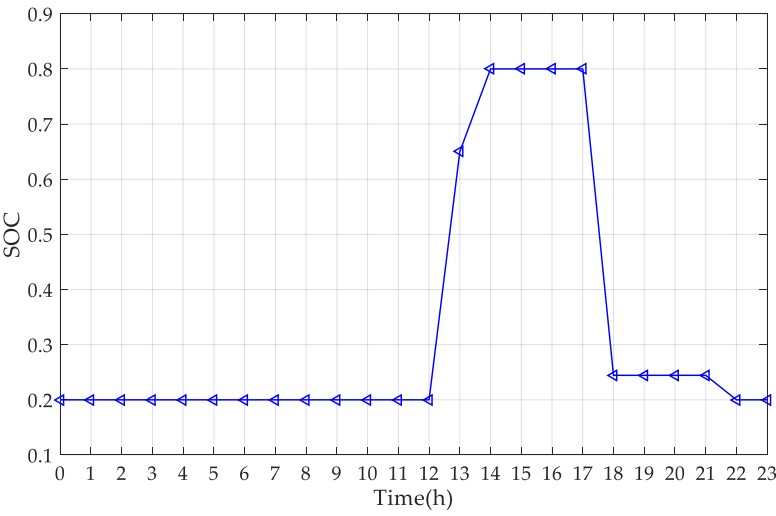

**Figure 15.** SOC curve of EES (scenario 3).

As can be seen from the above figure, the SOC of EES is within a limited range, meeting the requirements of the upper and lower limits of SOC, and the state of the beginning and end of the cycle is consistent.

The curtailed power before and after optimization are compared on a graph, and the scenario is analyzed in detail in relation to the changes in the curtailed power. The comparison of power curtailment is shown in Figure 16.

It can be seen from Figure 16 that from 2:00 to 5:00 on this typical day, since the curtailed power after the PSH operation has been completely accommodated, the EES only needs to be charged once and discharged once. The curtailed power reached the maximum value of 12,031.05 MW·h from 12:00 to 13:00. Due to the pumping of the PSH unit, the curtailed power dropped to 5961.05 MW·h. On this basis, the EES was charged for accommodation, and the curtailed power further decreased to 3961.05 MW·h.

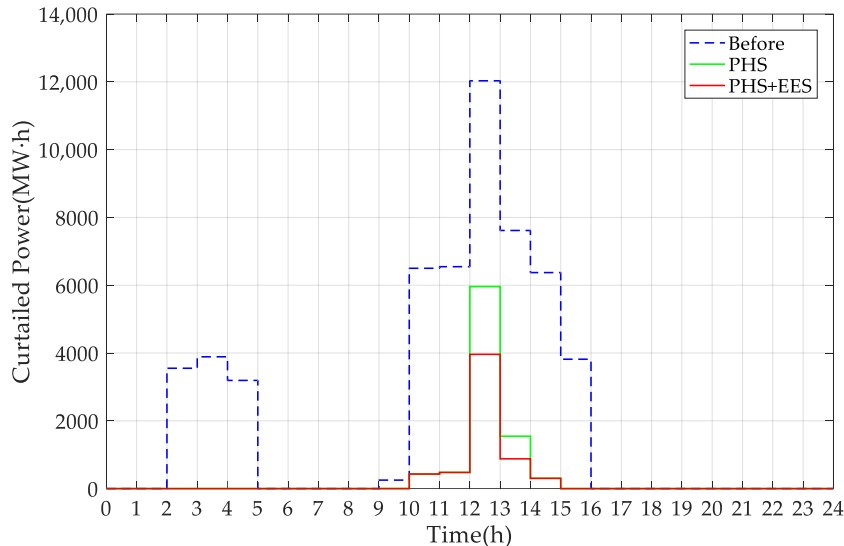

**Figure 16.** Comparison of power curtailment before and after optimization (scenario 3).

5.2.4. Configuration Optimization of EES

During the simulation cycle of this typical week, the operation of the EES is not considered, and the operation optimization of the PSH is carried out independently. The total amount of curtailed power before optimization is calculated from the original curtailed power, the total amount of curtailed power after adding PSH optimization is calculated from the optimization result, and the power curtailment rate is calculated from the renewable energy power generation, which is convenient for the next step of energy storage configuration optimization. The data are shown in Table 2.

**Table 2.** Optimization results of power curtailment and its rate.

| Term | Date |
|---|---|
| Total curtailed power before optimization (MW·h) | 385,760 |
| Total curtailed power after PSH optimization (MW·h) | 128,483 |
| Percentage of drop in curtailed power | 66.69% |
| Total renewable energy generation (MW·h) | 2,301,608 |
| Power curtailment rate | 5.58% |

After reducing power curtailment through PSH in the power grid, it may still fail to meet the requirements of the power curtailment rate. It is obviously more troublesome to expand the reservoir capacity of the PSH, but configuring EES can save costs and provide additional functions such as frequency regulation. Therefore, in order to achieve a further reduction in power curtailment and achieve the given power curtailment rate index, a certain capacity of the EES should be configured. In this scenario, it is necessary for PSH and EES to jointly accommodate renewable energy.

For the configuration optimization of EES, according to the objective function of the EES configuration and its constraints, the configuration of EES is solved based on the data of typical weeks. Among them, the expected power curtailment rate is set based on the data of the power curtailment rate in Table 2, and the rated power of EES that needs to be configured under different power curtailment rates is optimized. The configuration results of EES are shown in Table 3.

Table 3 shows the minimum cost of investment required to achieve the power curtailment rate target in various situations, which provides the power grid with recommendations for EES configuration. The relationship between the cost of investing in EES and the benefit of reducing power curtailment can be weighed according to needs. The above data are plotted to explore the relationship between the power curtailment rate and the investment cost of EES, as shown in Figure 17.

**Table 3.** Optimization results for the configuration of EES.

| Power Curtailment Rate | Configured Power of EES (MW) | Configured Capacity of EES (MW·h) | Investment Cost of EES (Million Yuan) |
| --- | --- | --- | --- |
| 5.58% | 0 | 0 | 0 |
| 5.3% | 673 | 1346 | 2557.4 |
| 5.1% | 1149 | 2298 | 4366.2 |
| 4.9% | 1665 | 3330 | 6327.0 |
| 4.7% | 2201 | 4402 | 8363.8 |
| 4.5% | 2955 | 5910 | 11,229.0 |
| 4.3% | 4426 | 8852 | 16,818.8 |

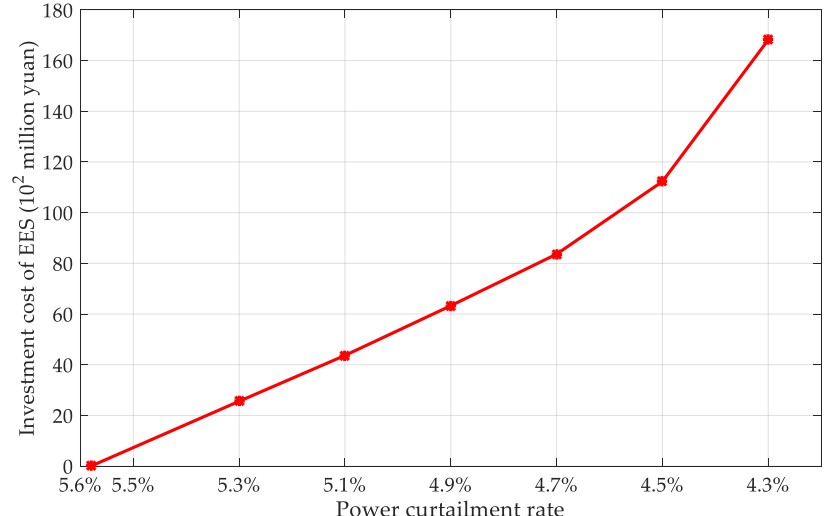

**Figure 17.** Investment cost of EES under different power curtailment rates.

It can be seen from Figure 17 that with the reduction of the power curtailment rate, it is necessary to configure larger capacity EES, which increases the investment cost. In addition, when the power curtailment rate is reduced to around 4.5%, the curve appears nonlinear, which means that when the power curtailment rate drops to a certain extent, to further improve the accommodation effect, it is necessary to configure incrementally larger capacity for the EES.

5.2.5. Involvement of Variable Speed Units

If all the units are fixed speed units, in the pumping condition, the units pumping water are in full power state, which results in more or less energy waste in certain periods. After replacing some fixed speed units with variable speed units, the power adjustment under pumping conditions is more flexible, and unbiased power compensation can be achieved in most time periods.

In this case, the two fixed speed units in PSH power station 3 are replaced with variable speed units of the same capacity, and the output range under pumping conditions is 70–100%. The simulation on the 7th day is used for comparison, and the comparison curves of the PSH operation after optimization on that day is shown in Figure 18.

It can be seen from the figure that after adding the variable speed units, the power under the pumping condition is significantly reduced. Specifically analyzing the optimization results, from 13:00 to 14:00, the original curtailed power during this period was 5560.15 MW·h. In the case of conventional units, the total pumping power of the PSH power station is 5690 MW; after adding the variable-speed unit, the total pumping power of the PSH power station is 5560.15 MW, which realizes the compensation of pumping power without deviation, which saves 129.85 MW of pumping power consumption compared with the conventional unit. Considering the total pumping power of the whole

day, without adding variable speed units, the total pumping power of the PSH power station on that day is 39,039.24 MW; after adding variable speed units, the total pumping power is 36,958.32 MW. A total of 2080.92 MW of pumping power was saved on that day. Considering the long-period power station operation, the variable speed unit will bring considerable energy savings, which verifies the necessity of adding the variable speed unit.

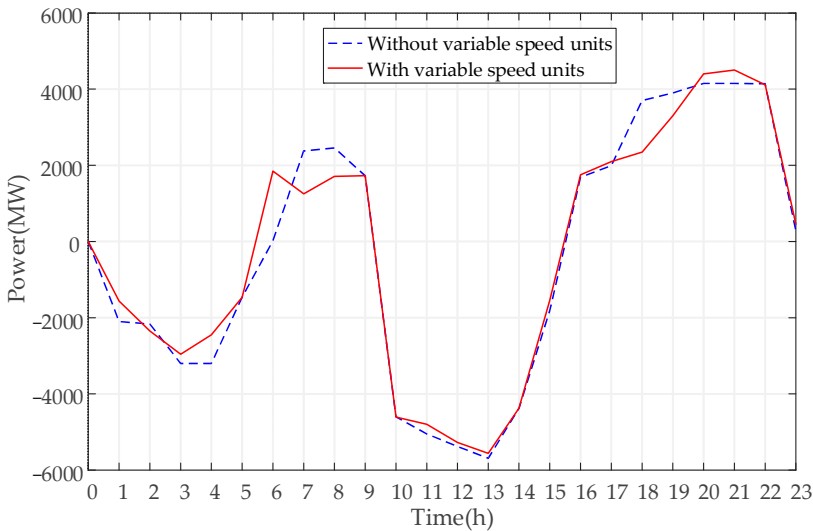

**Figure 18.** Comparison of PSH operation before and after adding variable speed units.

Comparing the total amount of curtailed power after optimization before and after adding variable speed units, it can be found that there is no difference in the reduction of the total amount of curtailed power on that day. This is because overcompensation without adding variable speed units will not affect the total amount of curtailed power after optimization. Therefore, it will not affect the configuration of EES. All in all, the variable speed unit can improve the flexibility of the unit combination, make the pumping compensation more accurate, and reduce the total power consumption of the PSH power station without changing the optimization effect. The comparative analysis in this part provides a basis for the appropriate use of variable speed units in the future construction of PSH power stations.

## 6. Conclusions

In this paper, aiming at improving the accommodation of renewable energy, a joint operation model of PSH and EES is proposed, and on this basis, the configuration of EES is studied.

In this paper, the proposed combined operation model of PSH and EES is used to reasonably arrange the operation of PSH and EES, so that the two cooperate to reduce the amount of curtailed power in the power grid and improve the accommodation of renewable energy. On this basis, using the proposed EES configuration optimization model, combined with the expected power curtailment rate, the corresponding EES configuration is solved, and the investment cost is calculated, which satisfies the actual situation for different needs for power curtailment rates. The rationality and effectiveness of the proposed optimization method and optimization model are verified by actual cases.

In the operation optimization of PSH, the influence of the participation of variable speed units on PSH operation is compared. Studies have shown that the addition of variable speed units will improve the flexibility of the unit's power output, make the pumping compensation more accurate, reduce the overall power consumption of PSH, and reduce energy waste. Therefore, the appropriate input of variable speed units should be considered in the construction of the actual PSH power station.

In future work, factors such as the hydraulic head will be considered to enrich the PSH model. The site selection optimization of EES will be studied considering the scenario

of the actual power grid. Additionally, the complementary of PSH and EES in terms of timescales, and start-stop costs, will be discussed more fully.

**Author Contributions:** Conceptualization, L.S. and F.Y.; methodology, L.S. and F.Y.; software, L.S., F.Y. and Y.L.; validation, L.S., F.Y. and T.Z.; formal analysis, L.S. and F.Y.; investigation, Y.L. and K.Y.L.; resources, F.W.; data curation, L.S. and F.Y.; writing—original draft preparation, L.S. and F.Y.; writing—review and editing, L.S., Y.L. and K.Y.L.; visualization, F.Y. and Y.L.; supervision, T.Z. and K.Y.L.; project administration, L.S. and F.W.; funding acquisition, T.Z. All authors have read and agreed to the published version of the manuscript.

**Funding:** This work was supported in part by State Key Laboratory of Smart Grid Protection and Control under Grant SGNR0000KJJS2200297 and in part by National Natural Science Foundation of China under Grant 52061635102.

**Institutional Review Board Statement:** Not applicable.

**Informed Consent Statement:** Not applicable.

**Data Availability Statement:** Not applicable.

**Acknowledgments:** We would like to extend special thanks to the editors and reviewers for their insightful advice and comments on the manuscript.

**Conflicts of Interest:** The authors declare no conflict of interest.

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
