# Peer review of "Optimal Configuration of Electrochemical Energy Storage for Renewable Energy Accommodation Based on Operation Strategy of Pumped Storage Hydro"

_sustainability, doi:10.3390/su14159713_

Round 1

Reviewer 1 Report

Pumped storage (ESP) is a technology that has been operating for many years for large-scale electricity storage. They convert this electrical energy into the potential gravitational energy of water. The process involves pumping water from the lower tank to the upper tank at a time when electricity production is greater than demand (for example at night), and then when electricity consumption is high (at peak times), the reverse process occurs. Pumped-storage power plants are a powerful battery (energy storage) with huge capacity and energy efficiency.

Hydroelectric power plants are concrete dams built on rivers, which are designed to stop and dam up water. It is collected in the retention reservoir and gradually discharged from it. This allows the water to move the turbine blades that drive the generator that produces electricity. Currently, around 16% of the world's electricity is generated by hydroelectric plants. All Renewable Energy Sources are aimed at reducing the consumption of natural resources, such as coal and oil, and reducing the emission of harmful substances into the atmosphere. The advantages of hydropower also include the improvement of the hydrological balance and better conditions for navigation. The construction of hydropower plants also helps regulate rivers and equalize flows, thus reducing the risk of flooding. In addition, hydroelectric plants provide jobs, which has a positive effect on the local economy. The disadvantages of hydroelectric power plants are mainly high construction costs and significant interference in the natural environment. The most common negative effects are silting of rivers and reservoirs, as well as contamination of groundwater and groundwater. Construction of a hydroelectric power plant is also associated with a significant transformation of the landscape and the displacement of the population.

Work, therefore, exhausts those and trends, and therefore accepts it fully.

Author Response

Dear reviewer:

We are very grateful to you for reviewing our manuscript titled "Optimal Configuration of Electrochemical Energy Storage for Renewable Energy Accommodation Based on Operation Strategy of Pumped Storage Hydro"(ID: sustainability-1838658).

We have carefully studied your comments on pumped storage and hydroelectric power plants and incorporated some of them into our manuscript. Thank you very much for accepting our manuscript.

Yours sincerely,

All authors

3 Aug., 2022

Reviewer 2 Report

Title: Optimal Configuration of Electrochemical Energy Storage for Renewable Energy Accommodation Based on Operation Strategy of Pumped Storage Hydro

·         The authors addressed few of the reviewer comments in the revised manuscript. The technical writing of the paper should be significantly improved. Please double check all sentences and make sure that they are all grammatically correct. What are the main objectives of the paper? They should be directly pointed out at the very beginning in the Abstract. Clearly highlight the novelty in the work presented and the authors fails to present incremental work to the existing works. No proper design considerations, it is very casual and unjustified. Similar types of analysis/works have already discussed in the previous literature related to the presented problem then what is new in this work?

·         Rewrite the Introduction section with highlighting the Research gap, motivation and author’s Contribution clearly.

·         The authors should exactly mention ratings of the renewable energy resources and storage devices considered in detail. Also show the power balance response of the entire system considered for the study. The main contributions of the paper should be clearly demonstrated in methodology and results section, further summarized.

·         Avoid numbering in the conclusion section and rewrite the section with including future work.

Author Response

Dear reviewer:

We are very grateful to you for giving us an opportunity to revise our manuscript. we appreciate you very much for your positive and constructive comments and suggestions on our manuscript entitle "Optimal Configuration of Electrochemical Energy Storage for Renewable Energy Accommodation Based on Operation Strategy of Pumped Storage Hydro"(ID: sustainability-1838658).

We have studied your comments carefully and tried our best to revise our manuscript according to the comments. For each of your suggestions and comments, we have responded in detail one by one and revised it in the manuscript. Please see the attachment. Thanks again for your hard work!

Yours sincerely,

All authors

3 Aug., 2022

Reviewer 3 Report

The manuscript reported the optimal configuration of EES combined with PSH, aiming to improve the accommodation of RES. An optimal configuration method of EES capacity is proposed and the simulation is carried out in the actual power grid. It can be concluded that by rationally configuring the capacity of EES, the desired power curtailment rate of the power grid can be achieved, and the necessity of configuring variable speed units is verified.

I consider the content of this manuscript will definitely meet the reading interests of the readers of the Sustainability journal. However, there are certain English spelling and grammar issues, and also the discussion and explanation should be further improved.

Therefore, I suggest giving a minor revision and the authors need to clarify some issues or supply some more experimental data to enrich the content. This could be comprehensive and meaningful work after revision.

Detailed comments can be found in the attached PDF file.

Author Response

(The authors gave the same response as above.)

Round 2

Reviewer 2 Report

The authors addressed most of the reviewer comments in the revised manuscript.